# Mechanical Behavior and Morphological Study of Polytetrafluoroethylene (PTFE) Composites under Static and Cyclic Loading Condition

**DOI:** 10.3390/ma14071712

**Published:** 2021-03-31

**Authors:** Karolina Mazur, Aneta Gądek-Moszczak, Aneta Liber-Kneć, Stanisław Kuciel

**Affiliations:** 1Faculty of Materials Engineering and Physics, Institute of Materials Engineering, Tadeusz Kosciuszko Cracow University of Technology, al. Jana Pawła II 37, 31-864 Cracow, Poland; mazur117@o2.pl; 2Faculty of Mechanical Engineering, Institute of Applied Informatics, Tadeusz Kosciuszko Cracow University of Technology, al. Jana Pawła II 37, 31-864 Cracow, Poland; aneta.gadek-moszczak@pk.edu.pl; 3Faculty of Mechanical Engineering, Institute of Applied Mechanics and Biomechanics, Tadeusz Kosciuszko Cracow University of Technology, al. Jana Pawła II 37, 31-864 Cracow, Poland; aliber@pk.edu.pl

**Keywords:** glass fiber, bronze particles, coke particles, graphite particles, hysteresis loops, compression test

## Abstract

The key goal of this study was to characterize polytetrafluoroethylene (PTFE) based composites with the addition of bronze particles and mineral fibers/particles. The addition of individual fillers was as follows: bronze—30–60 wt.%, glass fibers—15–25 wt.%, coke flakes—25 wt.% and graphite particles—5 wt.%. Both static and dynamic tests were performed and the obtained results were compared with the microscopic structure of the obtained fractures. The research showed that the addition of 60 wt.% bronze and other mineral fillers improved the values obtained in the static compression test and in the case of composites with 25 wt.% glass fibers the increase was about 60%. Fatigue tests have been performed for the compression-compression load up to 100,000 cycles. All tested composites show a significant increase in the modulus as compared to the values obtained in the static compression test. The highest increase in the modulus in the dynamic test was obtained for composites with 25 wt.% of glass fibers (increase by 85%) and 25 wt.% of coke flakes (increase by 77%), while the lowest result was obtained for the lowest content of bronze particles (decrease by 8%). Dynamic tests have shown that composites with “semi-spherical” particles are characterized by the longest service life and a slower fatigue crack propagation rate than in the case of the long glass fibers. In addition, studies have shown that particles with smaller sizes and more spherical shape have a higher ability to dissipate mechanical energy, which allows their use in friction nodes. On the other hand, composites with glass fiber and graphite particles can be successfully used in applications requiring high stiffness with low amplitude vibrations.

## 1. Introduction

Today, it can be observed that the interest in advanced composite materials has grown rapidly. The main reason for the growing popularity of polymer composites is their good mechanical properties, such as high specific strength and stiffness, low density, high fatigue strength, high damping and low thermal coefficient. Due to its excellent properties polymeric materials are widely used in the automotive and aerospace industries as functional materials. A polymer material with unique properties that is often used in engineering applications is polytetrafluoroethylene (PTFE), more commonly known under the trade name Teflon [1]. It is a hydrofluoric compound with a very stable main chain structure due to strong C–F chemical bonds. Due to the high molecular cohesion of PTFE chains, it is characterized by high chemical/corrosion resistance, low friction coefficient (below 0.1) and has a wide operating temperature range from −260 °C to 260 °C [2]. Whereas, due to low wear resistance, poor dimensional stability, high creep deformation, unmodified PTFE is used as a material for lubricants and seals (low friction and chemical inertness) [3]. However, modified PTFE is widely used in the automotive, electrical and electronics industries, aerospace, communications, construction, medical devices and implants, blood vessel prostheses, special packaging and protective clothing.

To extend the application of PTFE, it is more and more frequently modified. Mainly modifications of PTFE are intended to increasing the tribological properties of composites [4,5]. The most commonly used fillers for tribological applications are: carbon, bronze, glass fibers, graphite and molybdenum disulfide in varying percentages, and sometimes in combination, in order to improve wear resistance and coefficient of friction [6]. Glass fiber-filled composites reduce flow under load, reduce wear and improve sliding properties. Graphite-modified composites are compositions for sliding applications. The addition of bronze affects the electrical conductivity and thermal volume. While the composites with coke content are produced to reduce flow under load, increase hardness and thermal conductivity [7,8]. In the work of Conte et. al. the tribological properties of composites based on PTFE reinforced with 25 wt.% carbon fiber (25CF), 60 wt.% bronze (60B), 15 wt.% graphite (15G) and 25 wt.% glass fiber (25GF) were tested [6]. Research has shown that the content of hard fibers significantly improves the wear resistance of composites. The best results were achieved by CF composites because their specific resistance (friction energy/mass loss) was the highest, i.e., 15 times higher than that of unmodified PTFE. The authors attribute this dependence on the presence of hard carbon fibers (increased hardness). On the other hand, composites with 60 wt.% bronze were characterized by the lowest wear resistance (1.1 MJ/g).

Currently, there is a trend in modifying PTFE, in in order to increase its mechanical properties. There are not so many publications in the world literature (as in the case of tribological studies) concerning the description of the results of mechanical tests of PTFE composites reinforced with fibers/particles, which is also indicated by other authors [9]. One of the authors describing the influence of fillers on the mechanical properties of PTFE composites is Zhang, who investigated the influence of carbon (0.5 to 2 wt.%) and aramid (1 to 9 wt.%) fibers on abrasive wear and mechanical properties [10]. The addition of carbon fibers significantly lowered the tensile strength and the ability to deformation. However, depending on the fiber diameter, an increase in Young’s modulus by 70% (7 µm) and 137% (200 µm) was observed. A similar dependence was shown by the addition of aramid fibers, however, with the increase in the fiber content, the authors did not observe a deepening decrease in tensile strength, as was the case with elongation. The highest increase in the modulus of elasticity was observed in composites with the addition of 5 wt.% aramid fibers, which was 1.8 times higher than in the case of unmodified PTFE. In another work, Vasiliev et al. described the influence of the low content of carbon fibers (1–5 wt.%) on the tribological and mechanical properties of composites [11]. Additionally, 1 wt.% kaolin was added to each composition. As in the previously described works, the addition of hard carbon fibers increased the wear resistance of PTFE composites. As the fiber content increases, the wear resistance increases more than 1000 times. On the other hand, even a small addition of fibers lowered the deformation capacity of composites, thus, causing an increase in compression stress at 10% strain from 16 MPa to 24 MPa.

In recent years, the use of polymer composites has significantly expanded, mainly in engineering applications where cyclical loads often occur (e.g., structural or load-bearing elements). In the case of polymeric materials, failure of the material occurs with less stress than in the case of normal static mechanical load [12]. On the other hand, for polymer composites reinforced with fibers/particles, fatigue occurs through fiber tearing, matrix fracture, matrix-fiber detachment, delamination, etc. Therefore, determining fatigue mechanisms is an important issue when determining the durability of polymeric materials. In general, fatigue testing of polymer composites is not as often a subject of research as static testing, mainly due to the complicated methodology and long-term testing. As for composites based on PTFE, there is a low number of researches determining the fatigue life [13]. Considering the growing interest in PTFE composites in engineering applications, it is justified to test these composites in terms of fatigue testing.

In the case of composites reinforced with particles/fibers, their initial mechanical, thermal or physical properties are important, as well as their morphology, which is often crucial for the initial properties of polymer composites [14]. One of the methods of assessing the influence of particle geometry is the finite element method. This method was used by Qing to investigate the effects of particle size, locations and orientations on mechanical properties of metal-matrix composites [15]. SiC particles of various shapes (circular, octagonal, hexagonal and square) were used as the filler. As shown by the tensile failure stresses results, they are dependent on the structure of the filler, and the best results were obtained in composites with circular particles, and the lowest with square ones. Another work shows the influence of particle geometry on the properties of composites was Lebar et al. [16]. In their work, they investigated the influence of the particle shape on the spall strength of polyurethane composites reinforced with aluminum oxide. As the research showed, not the shape of the particles but the particle/matrix adhesion had a decisive influence on the properties. Kumar et al. investigated the influence of the shape of carbon nanofillers (s(CB, CNT and GR)) on the mechanical and electro-mechanical properties of resin-based composites (RTV silicone) [17]. The particles were characterized by different morphology: CBs had an oval shape and formed agglomerations, CNTs were characterized by an elongated shape with a large specific surface area, while GR was similar to platelets, but agglomerated to a lesser extent than CB. The research showed that the highest properties were characterized by composites reinforced with elongated fillers (CNT) and the increase in Young’s modulus was 350%, while the lowest values were obtained for composites with CB (increase by 130%). This was most likely due to particle agglomeration and the inability to transfer the load from the die to the filler.

In the presented work, PTFE-based composites were produced by pressing. Bronze powder and mineral fibers/particles were used as a reinforcement element. The produced materials were subjected to mechanical and fatigue tests. To analyze the results more precisely, the images of breakthroughs after static and dynamic tests were analyzed. Moreover, a quantitative analysis of the microstructure of composites (before the tests, after static tests and after dynamic tests) was performed with the use of computer software. To the best of the authors’ knowledge, such a comprehensive characterization of PTFE-based materials has not been performed previously. The proposed a research methodology, which took into account the static and dynamic tests, with the simultaneous comparison with scanning electron microscopy (SEM) images, will not only allow to determine the parameters of materials, but also allow to understand the mechanisms occurring during the tests, taking into account various loads. This approach will help to expand the use of PTFE composites.

## 2. Materials and Methods

### 2.1. Materials

In this study PTFE in powder form produced by Zakłady Azotowe in Tarnów (Poland) under the trade name of Tarflen was used as matrix. Four types of fillers were used as reinforcement: bronze powder—Cu90Sn9Zn1-2 (Bimex, Gdańsk, Poland), glass fibers (Krosglass S.A., Krosno, Poland), coke flakes and graphite particles (ZEW, Raciborz, Poland). In Table 1 abbreviations and compositions of produced composites were presented.

Rectangular samples with dimensions: 50 × 12.5 × 12.5 were prepared for the tests by pressing in hydraulic presses. The mixed matrix and fillers were dried at 60 °C for 24 h, followed by pressed at 10–70 MPa at ambient temperature. The specimens were removed from the mold and were relaxed at ambient pressure and temperature for approximately 12 h prior sintering. The products formed in the pressing process, removed from the molds or in the molds, were sintered in furnaces with internal air circulation, at a temperature of 360–380 °C with a rate of 40 °C/h. The sintering process, depending on the weight of the product, takes up to 40 h. Thereafter, the sintered products were cooled to room temperature in the oven at 40 °C/h. In Figure 1, a schematic diagram of the manufacturing process was shown.

### 2.2. Methods of Testing

The samples were tested in cyclic compression condition on universal testing machine INSTRON 8511.20 (Instron, Norwood, MA, USA) at the speed of increasing the force −1 kN/min at ambient temperature. During cycling compression test, mechanical hysteresis loops were recorded in the force-shortening system, and the computer program calculated the mechanical energy dissipated in each cycle, the secant modulus of elasticity and the value of the average deformation (by assessing the change in position, the mean hysteresis loop) as a function of time. The tests were performed at a frequency of 10 Hz, for 100,000 full load cycles. The dissipation energy was calculated on the basis of the surface area of the hysteresis loops obtained in one complete strain cycle. The Mathcad Prime 6.0.0.0 (PTC, Boston, MA, USA) program was used for the calculation.

In addition, static tensile and compression tests was performed by using universal MTS Criterion Model 43 (MTS Systems Corporation, Eden Prairie, MN, USA) testing machine according to ISO 527 at ambient temperature. The crosshead speed for tensile tests was 2.5 mm/min.

To exclude errors and create statistical analysis, at least three samples were tested for each test and for each produced material.

In order to observe and characterize the morphologies of PTFE composites scanning electron microscope (SEM) (JSM–5510LV, JEOL, Tokyo, Japan) in low-vacuum at 20 vK was utilized. Images of breakthroughs after static tensile, static compression and dynamic compression tests were taken.

The quantitative assessment of the microstructure of the tested composites was carried out with the Aphelion v. 3.2 software (ADCIS, Saint-Contest, France). The algorithm described by Wojnar et al. was used [18]. The analysis was made on images at a magnification of 500, both longitudinally and transversely, and involved at least 30 particles. An analysis of the fillers was performed, which determined the following parameters: volume fraction of particles in the matrix, particle surface area and Feret diameters F(0) and F(90) based on which the shape coefficient was calculated. The shape coefficient (aspect ratio) is defined by the ratio of the F(90) to the F(0). Due to the lack of differences in the method of measurement, the shape coefficient calculated on the basis of transverse measurements was provided for the analysis.

The obtained results were related to the results obtained from the static and dynamic compression tests.

## 3. Results and Discussion

### 3.1. Characteristics of Fillers

Figure 2 presents images showing the morphology of particles used as fillers in the produced composites. As can be seen, the particles have different structures. Bronze particles have a globular structure with dimension in the range of 20–30 µm, and their surface is not as developed as in the case of coke and graphite particles. The coke and graphite particles have a developed lamellar structure of quite irregular shape. The graphite particles are much larger and much more elongated than the coke particles. In contrast, glass fibers have the typical elongated structure characteristic of fibers. The diameter of the glass fibers is 11 µm and the length is max. 200 µm.

### 3.2. Static Tests

The impact of different types and various contents of fillers on the mechanical properties (tensile and compressive tests) of PTFE composites is shown in Table 2. The highest level of deformation was characterized by unmodified PTFE—416%. This indicates the flexible nature of the material, where the molecular chain retains higher integrity under load. Moreover, as the amount of filler increases, the deformability decreases—more inclusions resulted in more matrix discontinuities.

For composites with the addition of bronze powder (30–40 wt.%), no significant differences were noticed in the static tensile test, especially for the breaking stress value. Only the addition of a high amount of filler (60 wt.%) caused a decrease in breaking stress by about 30% and elongation by 57%. This phenomenon is most likely related to the cracking mechanism (microcracks may appear at the interface between bronze powder and PTFE, which leads to their slow propagation) and the exhaustion of the possibility of transferring elastic strains through the PTFE matrix, for high levels of filling with much stiffer fillers. While the addition of bronze powder up to 40 wt.% did not adversely spill over into breaking strength and deformation, the addition of mineral fillers (glass fiber, coke and graphite) decreased this result by at least 40% (PTFE/15GF). The highest decrease was observed for composites with coke flakes—60% and 76% for breaking stress and elongation, respectively. These decreases are due to disturbances caused by the addition of fillers, which mechanically impeded the movement of the PTFE chains and reduced the continuity/integrity of the matrix [19]. The higher the filler content, the lower the deformability. This relationship indicates the formation of discontinuities in the matrix, which reduce the plastic deformation capacity of the tested composites. The addition of particles resulted in the creation of more places where breaking stress could occur. The higher the volume of a given filler, the higher the number of particles and thus the more destructive places. These studies also confirm this relationship: PTFE/25C (29 vol.%—100%) < PTFE/60B (29 vol.%—181%) < PTFE/25GF (23 vol.%—200%) < PTFE/25C (15 vol.%—200%) < PTFE/15GF (14 vol.%—250%) < PTFE/40B (16 vol.%—367%) < PTFE/40B (11 vol.%—383%). In the case of tensile strength, the obtained relations are caused, due to the very stable F bonds with the C–C bond, PTFE hardly reacts with other compounds so there are only simple physical bonds between the fiber and the matrix. This leads to poor filler/matrix adhesion, which makes it impossible to transfer loads from the matrix to the fiber [20]. Similar values of deformation decrease were observed in the work of Zhang et al. where the addition of 15 wt.% glass fiber led to a decrease in the deformation value to the level of 221, 15% [10].

Comparing the results obtained in the tensile and compression tests (Table 2), it can be seen that the values obtained during the compression test are higher than in the tensile test. This may indicate a certain anisotropy and higher compressive strength of PFTE composites due to the significant flowability of the polymer matrix. As the bronze powder content increases, Young’s modulus increases (PTFE/30B—decrease by 30%; PTFE/60B—increase by 26%). This indicates that, in the case of bronze powder, a minimum of approximately 40 wt.% is needed to increase the stiffness of the material. It is worth noting that in other studies, where the addition of bronze was about 40 wt.% (38.5 wt.%), the modulus of elasticity under static compression was much lower –369.7 MPa [1]. With the increase in the volume of the filler, the Young’s modulus increases under compression, the highest increase was recorded for PTFE/25GF (23 vol.%)—69%, and the lowest for PTFE/30B (11 vol.%)—a decrease by 30%. Due to the high stiffness and larger dimensions of glass fiber to bronze and coke particles, despite the higher content (29 vol.%), glass fiber composites have a higher Young’s modulus due to the above-mentioned aspects. In addition, from the research it can be concluded that larger particles with elongated shape caused a higher increase of Young’s modulus: PTFE/15G—802 MPa; PTFE/25GF—1291 MPa, where for the content of 25 wt.% of smaller less elongated coke flakes Young’s modulus was 952 MPa.

As the bronze powder content increases, both the force at 2% and 15% permanent deformation increases, which indicates an increased ability to transfer compressive loads. On the other hand, the addition of carbon fillers improved all the values obtained during the static compression test. The optimal values were found for composites with 25 wt.% coke flakes content, where the improvement was 24%, 45% and 66% for Young’s modulus, force at 2% and 15% permanent deformation, respectively.

### 3.3. Dynamic Tests

Knowledge of the dynamic properties of reinforced polymer composites is important when considering energy dissipation processes in applications with cyclic loading. Viscoelasticity is a typical behavior of polymeric materials and also plays an important role in polymer composites. Figure 3 shows the results obtained during the cyclic compression test of PTFE composites. Based on the recorded hysteresis loops, the energy dissipated in each cycle, the secant modulus of elasticity and the values of the average strain as a function of time were calculated. All tested composites show a significant increase in the modulus as compared to the values obtained in the static compression test. This value stabilizes after about 2000 cycles and shows only a slight downward trend with the passage of time and the development of fatigue processes. The addition of bronze in the amount of 30–40 wt.% did not significantly change the modulus compared to the unmodified polymer. The highest increase was observed for PTFE/25GF composites, obtained value—4.8 GPa. The addition of both 15 wt.% glass fibers and graphite particles resulted in similar values of the elasticity modulus of approx. 4 GPa. We observed a similar tendency in our previous research, where Poly(oxymethylene) (POM) with the addition of glass and carbon fibers (content from 0 to 40 wt.%) was tested [21]. As the content of glass fibers increases, the Young’s modulus obtained during cyclic loading increases (for content of 20 wt.% of fibers the Young’s modulus was about 3 times higher than for neat POM).

A similar change is shown by the average strains presented in Figure 3c,d. Initially, there is an abrupt increase in strains, which stabilizes and remains constant as the number of cycles increases. The higher the mass fraction of the filler, the higher the dimensional stability during long-term loads changing with time. Composites with the addition of glass fiber and 60 wt.% bronze powder were characterized by the highest dimensional stability, while composites with 30 wt.% bronze powder were the lowest.

Figure 4 shows the change in energy dissipated for composites based on PTFE. The addition of bronze powder to the PTFE matrix improves the ability of such composites to dissipate mechanical energy, possibly by increasing their thermal conductivity with increasing amounts of filler, which confirms the desirability of their application in friction junctions. Unmodified PTFE has the ability to increase its ability to dissipate mechanical energy, as the number of cycles increases, the amount of energy dissipated in each of them increases to the tested level of 100,000 cycles. This is probably due to the high adaptability of this material under loads not exceeding the compressive strength associated with the technology of its production, consisting in pressing and sintering the PTFE suspension, resulting in a microporous structure capable of dissipating mechanical energy. In the case of a significant 60% mass fraction of the filler, the ability to dissipate mechanical energy, although initially much higher, begins to decrease with the development of fatigue processes in the composite caused by the progressive decohesion of the composition components. Interestingly, for the various tested levels, the amount of energy dissipated after 100,000 cycles begins to oscillate towards a constant level, which may indicate the viscoelastic heating of the matrix and the dominance of this type of failure mechanism in the fatigue process of such composites.

The addition of chopped glass fiber causes a decrease in the ability to dissipate mechanical energy, while reducing, more than graphite or coke, the natural creep tendency of PTFE, especially under long-term loads, and increasing the values of modulus of elasticity, especially at a higher degree of filling. Moreover, in the case of fibrous fillers, the higher the fiber content, the lower the resistance to fatigue crack propagation, which is also confirmed by these tests [22]. Due to the high stiffness of the glass fiber, it is difficult to expect a significant improvement in the ability to dissipate energy by PTFE composites with glass fiber (lower viscoelastic deformation ability). In addition, the adhesion between the fiber and the matrix was not sufficient and the fibers became drawn under cyclic loads. A similar relationship was observed by Aglan et al. They investigated the effect of cyclic loads (tension-tensions) on PTFE composites with 15 and 25 wt.% GF and 15 wt.% G [22]. They showed that the longest fatigue life was characterized by unmodified PTFE (1.5 million cycles), and the lowest for composites with 25 wt.% GF (85,000 cycles). In addition, the authors of the study summed up, that the particulate filler tends to reduce the fatigue cracking resistance of PTFE, more than the short fibrous filler at the same content. Additives in the form of fibers tend to build ‘bridges’, which makes the propagation of cracks easier and faster compared to spherical particles. This situation has not been significantly observed in the presented work. Most likely, due to the type of load applied, being compression-compression and in the case of a literature source it was tensile-tensile.

### 3.4. Morphological Analysis of Composites

For a more complete analysis of phenomena occurring in composites during the tests, the compression and fatigue sections of the sample were observed by SEM. Unmodified PTFE is characterized by a laminar structure with a terraced (layered—marked with red arrows in Figure 5) arrangement. The average length of the lamellas was about 1 µm, and the thickness of the single crystals was about 0.01 µm. The lamellar structure of PTFE composites has already been described in the literature [23]. Cyclic loading changes the crystallographic structure of composites compared to static compression. The fracture surfaces subjected to cyclic loads have a less developed fracture surface compared to the fracture in static compression, which proves the degradation of the typical fine-crystalline structure of the spherulite.

As can be seen in Figure 6, Figure 7 and Figure 8, the addition of bronze powder only slightly changed the crystallographic structure of PTFE, while the type of the applied load (static or cyclic) changed this structure. In Figure 6, we can observe bronze particles with black rings around, what proves that the adhesion of the particles to the matrix is not sufficient. Additionally, voids can be seen which show that the bronze particles detached easily from the matrix. While at low filler contents we do not see microfibers, at higher contents they start to appear especially during cyclic loads. The higher the filler content, the more space for fibril nucleation. Due to the previously described chemical structure of PTFE, which practically prevents the formation of chemical bonds with other compounds, it can be said that the resulting microfibrils counteract PTFE against tearing [23]. This essence is confirmed by mechanical tests which show that with the increase of the filler, the ability of composites to plastic deformation decreases. It should be emphasized that in the case of compression tests, no significant formation of microfibrils is observed, but according to the literature sources, such a structure is visible primarily in tensile tests [24].

Figure 9, Figure 10, Figure 11 and Figure 12 show cross-sections of PTFE composites with mineral fillers (glass fiber, coke and graphite particles). The introduction of 15 and 25 wt.% filler into PTFE in the form of a short glass fiber stabilizes the growth of spherulites (Figure 9 and Figure 10). Moreover, uniform distribution of the reinforcement and poor adhesion with the matrix are observed. It should be presumed that the high dispersibility of the internal structure of spherulites is not affected by glass fiber. Due to the low cohesion of the fiberglass-matrix interface, these regions will be favored for nucleation of cracks during their deformation. The formation of dendritic structures (bridges) has also been observed in the literature for fibrous fillers such as glass fiber or carbon fiber [11]. The authors emphasize that the interactions between the polymer and the fiber took place locally, which confirms the very difficult interaction of PTFE with other materials.

Nevertheless, at high magnification, the observation of individual glass fibers indicates that during deformation PTFE bridges are formed between fibers and the matrix, which make it difficult to “pull out” individual fibers. Due to the large difference between the size of the spherulite and the filler, glass fiber in the used amounts does not significantly affect the size of the spherulites and does not affect the fine dispersion of the matrix of the composite. For the above-mentioned reasons, composites with glass fibers show the highest compressive strength (high hardness and strength of the filler contribute to this phenomenon).

In the case of small particles such as coke particles, their random distribution in the matrix was observed (Figure 11). Due to their lamellar structure, which is shown in more detail in Figure 2, and similar dimensions to matrix spherulites, coke plates favor the growth of matrix spherulites. As with other composites, microfibrils appear. As with glass fiber composites, there is a dendritic bond which indicates that there is a bond between the fiber and matrix. A confirmation of the relatively good adhesion of coke to PTFE is its relatively high compressive strength described before.

Figure 12 presents SEM photos of PTFE composites with the addition of 15 wt.% graphite. The addition of a plate-like filler (Figure 2) resulted in a significant reduction of the matrix spherulites. The flake arrangement of graphite (sometimes arranged parallel to each other) favors the spread of cracks on their surfaces. As in the case of the above-mentioned fillers, they do not affect the internal structure of spherulites. Spherulites are slightly smaller than the base material, and in the micro scale they have the same disc and plate structure (with a parallel arrangement of single crystals). This dense packing of the composite matrix hinders the development of the final crack. As a result of this phenomenon, composites filled with graphite show high resistance to compression (thus eliminating the unfavorable effect of the filler).

For a more detailed analysis of the influence of the type/size of particles on the properties of composites on the PTFE matrix, computer image analysis was performed on the basis of SEM images. The analysis was performed for three states: before the tests, after the static compression test and after the dynamic compression test. Figure 13a–c show the geometrical parameters of the particles, i.e., the Feret diameter (F0) for transverse and longitudinal measurements and the shape coefficient based on Feret minimum and maximum diameter [25]. In the analysis of the results, three groups of particle structure should be distinguished: a globular shape (bronze particles), a fibrous shape (glass fibers) and a plate-shape (coke and graphite particles). More precisely, the shape of the individual fillers is shown in Figure 2.

For the regular shaped particles (bronze, coke and graphite), there were no significant differences between the Ferret diameter measured both longitudinally and transversely. Larger differences appeared in the composites filled with fibers, because, as shown in the SEM photos (Figure 9 and Figure 10), the filler was not oriented in one direction. Additionally, the regular shape of bronze particles is confirmed by the results of the shape coefficient. The value of shape coefficient for bronze composites was closer to 1 than for glass fiber composites. Such relations are described in the literature sources, which indicate that if the shape coefficient based on Feret diameters is close to 1, its shape is similar to a sphere (isotropic area), while when it is close to 0, there is a fibrous structure (elongated area) [26].

Table 3 presents the results of the stereological analysis: The number of particles and the mean area of zones of influence. The data show that the cyclic loading affects the fragmentation of the fillers, which causes an increase in the number of particles per unit area. Moreover, as the amount of filler increases, the difference becomes more pronounced. The exceptions are composites with 15 wt.% glass fiber, which is probably caused by a calculation errors resulting from the parallel arrangement of the fibers to the cross-section of the sample. Additionally, the research shows that it is possible to find a correlation between the stereological indexes and the type and state of the composite stresses. Such a relationship was observed for composites with the addition of bronze powder, where it can be seen that for a low filler content, the number of particles is a more sensitive parameter for the type of input, and for large areas of the influence zone (Figure 14).

## 4. Conclusions

Modifications of highly crystalline fluorine material (with unique properties resulting from the shielding of the carbon chain by fluorine atoms) by pressing with bronze powder, glass fiber and mineral particles are primarily aimed at increasing the thermal conductivity in applications for sliding elements. Based on the obtained results the following conclusions can be drawn:The addition of fillers had a positive effect on the obtained values during the static compression test: force at 2% and 15% of deformation increased from 3% (PTFE/30B) to 66% (PTFE/25C) and Young’s modulus increased for composites with mineral fillers max. by 69%;Larger particles with elongated shape caused a higher increase of Young’s modulus: PTFE/15G—802 MPa; PTFE/25GF—1291 MPa, where for small bronze particles—534 MPa;As the volume of the fillers content increased, the tendency for plastic deformation decreased;Mechanical energy dissipation tests for high stress levels (100,000 cycles) have shown that pure PTFE has the surprising ability to increase its ability to dissipate mechanical energy with increasing number of cycles.The addition of bronze powder to PTFE further improves the ability of such composites to dissipate energy with increasing mass fractions of the filler. This confirms the possibility of their use in friction junctions.The addition of chopped glass fiber reduces the natural creep tendency of PTFE more than graphite or coke, especially under long-term loads, while increasing the values of their modulus of elasticity and compressive strength for such composites.

SEM microscopic images showed that insufficient fiber/matrix adhesion was observed, which contributed to a reduction in the properties obtained during the tensile test. In order to improve the results obtained in this work in the future, the surfaces of the particles should be modified or their fragmentation increased. Additionally, a compatibilizer may be considered to increase the miscibility of the apolar PTFE with the fillers.

## Figures and Tables

**Figure 1 materials-14-01712-f001:**
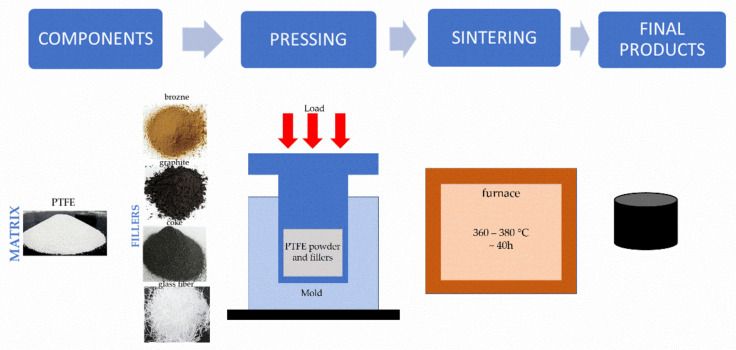
A schematic diagram of the manufacturing.

**Figure 2 materials-14-01712-f002:**
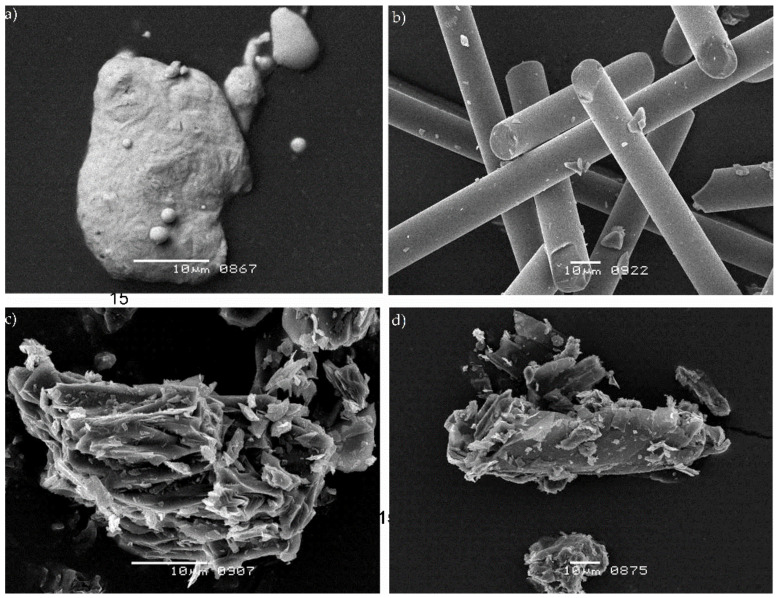
Typical shapes of fillers used in polytetrafluoroethylene (PTFE) composites: (**a**) bronze, (**b**) glass fibers, (**c**) coke and (**d**) graphite.

**Figure 3 materials-14-01712-f003:**
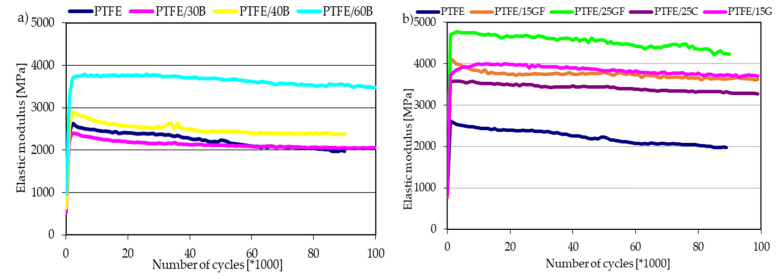
Modules of elasticity (**a**,**b**) and deformation (**c**,**d**) for PTFE and its composites measured during fatigue tests.

**Figure 4 materials-14-01712-f004:**
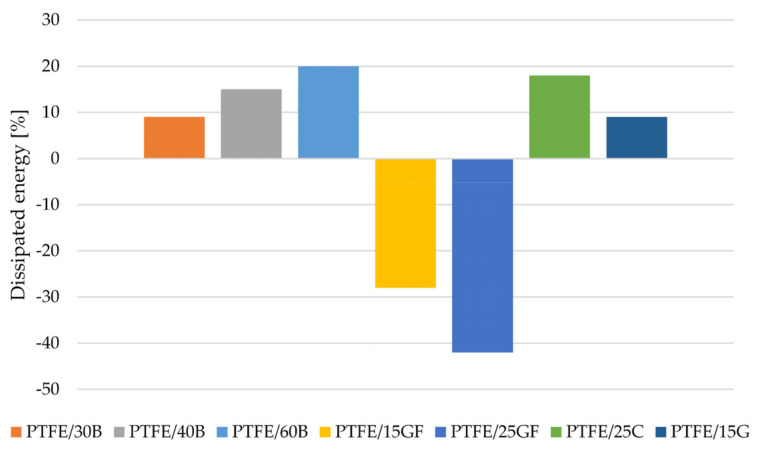
Change of the dispersion energy of composites based on PTFE.

**Figure 5 materials-14-01712-f005:**
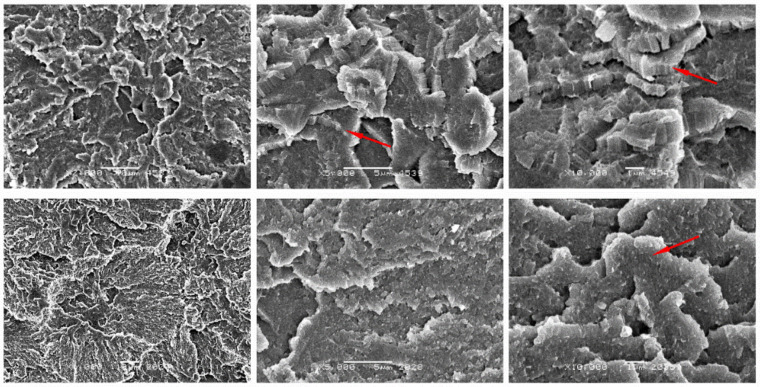
Scanning electron microscope (SEM) images of neat PTFE after static compression test (**top**) cyclic compression test (**bottom**) at different magnifications: ×2000, ×5000 and ×10,000.

**Figure 6 materials-14-01712-f006:**
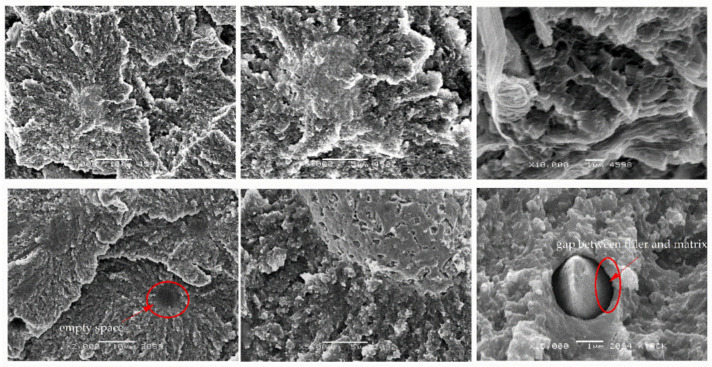
SEM images of PTFE/30B after static compression test (**top**) cyclic compression test (**bottom**) at different magnifications: ×2000, ×5000 and ×10,000.

**Figure 7 materials-14-01712-f007:**
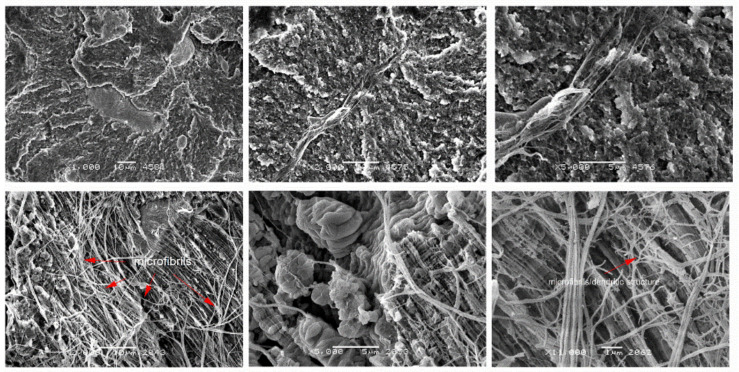
SEM images of PTFE/40B after static compression test (**top**) cyclic compression test (**bottom**) at different magnifications: ×2000, ×5000 and ×10,000.

**Figure 8 materials-14-01712-f008:**
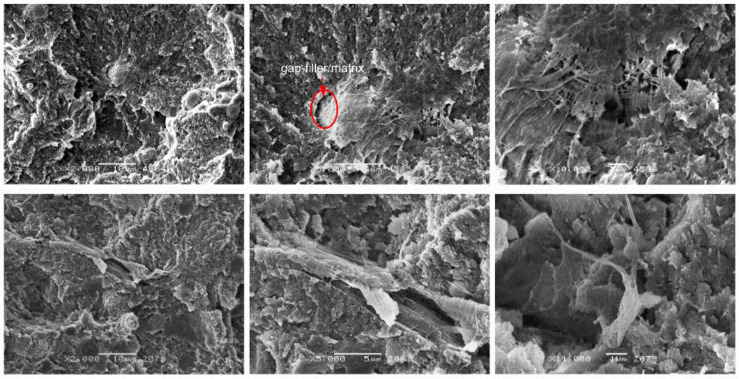
SEM images of PTFE/60B after static compression test (**top**) cyclic compression test (**bottom**) at different magnifications: ×2000, ×5000 and ×10,000.

**Figure 9 materials-14-01712-f009:**
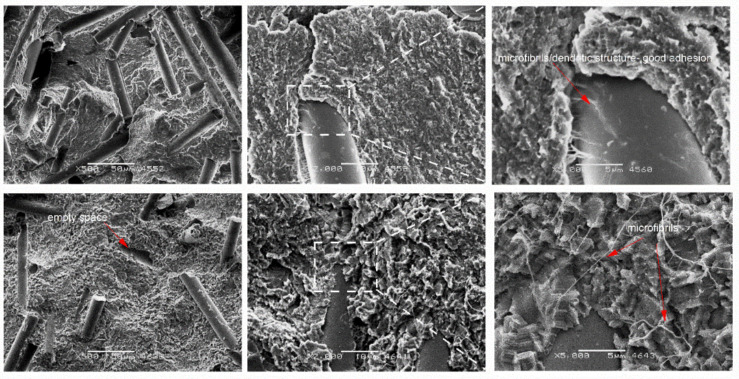
SEM images of PTFE/15GF after static compression test (**top**) cyclic compression test (**bottom**) at different magnifications: ×500, ×2000 and ×5000.

**Figure 10 materials-14-01712-f010:**
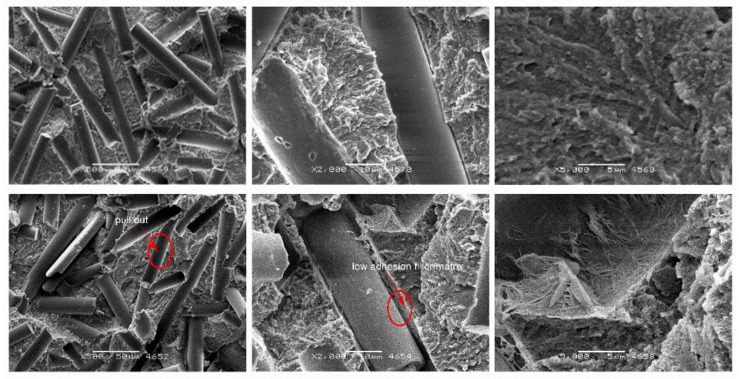
SEM images of PTFE/25GF after static compression test (**top**) cyclic compression test (**bottom**) at different magnifications: ×500, ×2000 and ×5000.

**Figure 11 materials-14-01712-f011:**
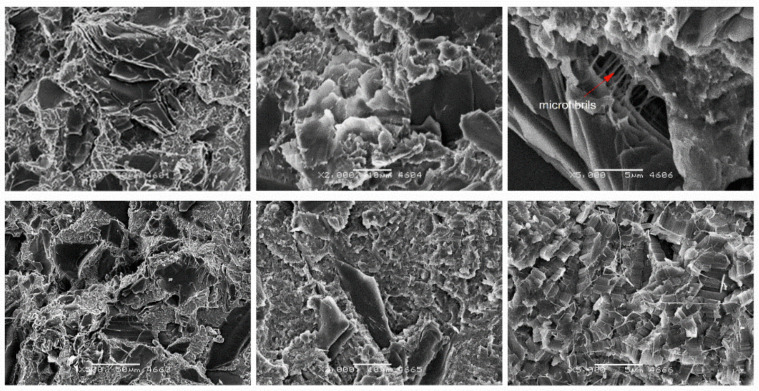
SEM images of PTFE/25C after static compression test (**top**) cyclic compression test (**bottom**) at different magnifications: ×500, ×2000 and ×5000.

**Figure 12 materials-14-01712-f012:**
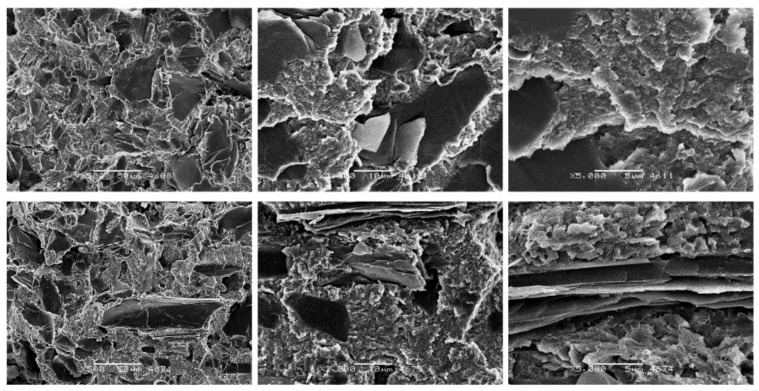
SEM images of PTFE/15G after static compression test (**top**) cyclic compression test (**bottom**) at different magnifications: ×500, ×2000 and ×5000.

**Figure 13 materials-14-01712-f013:**
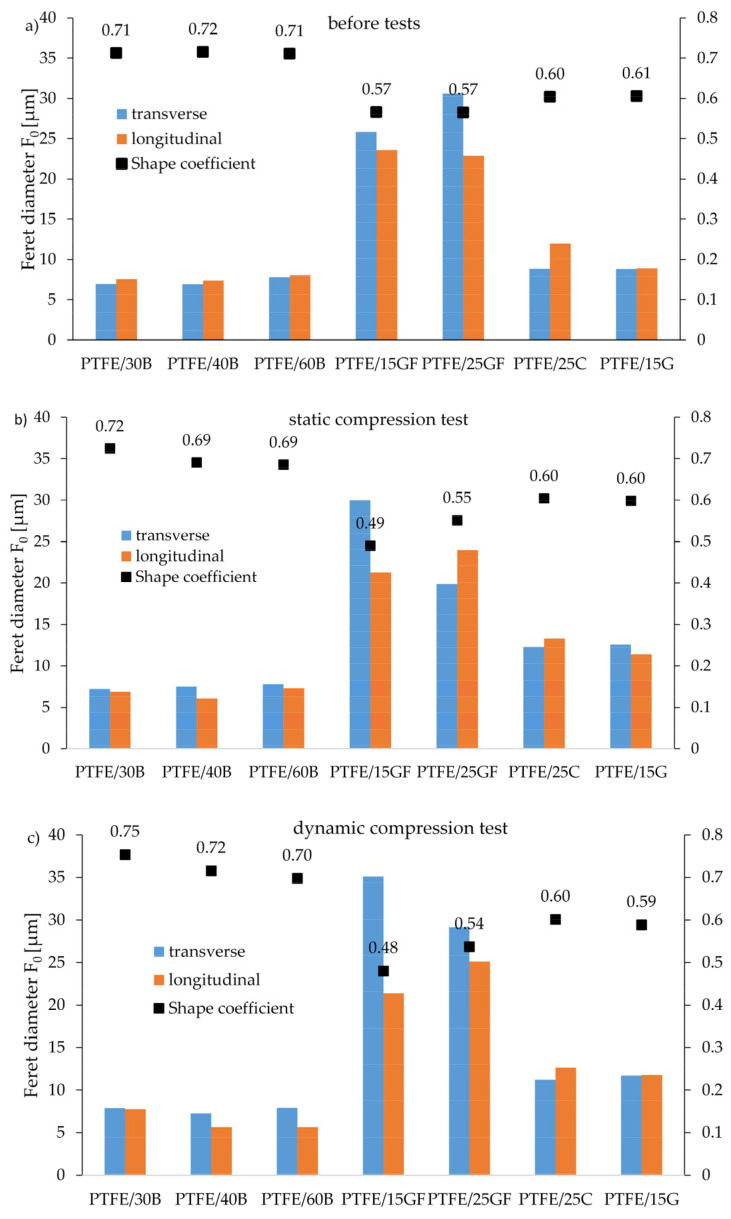
Comparison of Feret diameters for PTFE composites: (**a**) before tests, (**b**) after static compression tests and (**c**) after dynamic compression tests.

**Figure 14 materials-14-01712-f014:**
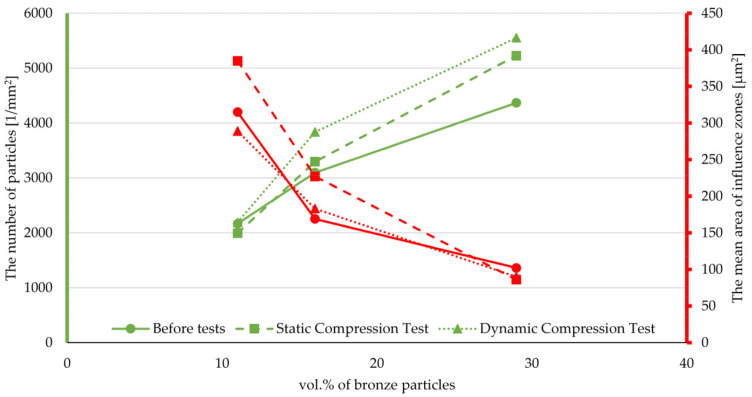
Changes in the number of particles and the mean area of the influence zones for increasing mass fractions of bronze powder in composites based on PTFE matrix.

**Table 1 materials-14-01712-t001:** Abbreviations and compositions of produced composites.

Abbreviation	Composition
PTFE	100 wt.% polytetrafluoroethylene
PTFE/30B	70 wt.% polytetrafluoroethylene + 30 wt.% (11 vol.%) bronze particles
PTFE/40B	60 wt.% polytetrafluoroethylene + 40 wt.% (16 vol.%) bronze particles
PTFE/60B	40 wt.% polytetrafluoroethylene + 60 wt.% (29 vol.%) bronze particles
PTFE/15GF	85 wt.% polytetrafluoroethylene + 15 wt.% (14 vol.%) glass fibers
PTFE/25GF	75 wt.% polytetrafluoroethylene + 25 wt.% (23 vol.%) glass fibers
PTFE/25C	75 wt.% polytetrafluoroethylene + 25 wt.% (29 vol.%) coke flakes
PTFE/15G	85 wt.% polytetrafluoroethylene + 15 wt.% (15 vol.%) graphite particles

**Table 2 materials-14-01712-t002:** Results from tensile and compression tests for PTFE and its composites.

Type	Static Compression Test			Static Tensile Test		
	Young’s Modulus, MPa	Force at 2% Deformation, kN	Force at 15% Deformation, kN	Breaking Stress, MPa	Strain at Break, %	Breaking Stress, MPa
PTFE	766 ± 29	2.22 ± 0.10	3.38 ± 0.14	30.0 ± 1.2	416 ± 20	766 ± 29
PTFE/30B	534 ± 23	2.28 ± 0.09	3.25 ± 0.13	39.9 ± 1.7	383 ± 17	534 ± 23
PTFE/40B	634 ± 28	2.55 ± 0.11	3.75 ± 0.16	32.4 ± 1.4	367 ± 15	634 ± 28
PTFE/60B	963 ± 45	3.15 ± 0.13	4.80 ± 0.20	22.0 ± 0.9	181 ± 8	963 ± 45
PTFE/15GF	781 ± 37	2.68 ± 0.11	4.11 ± 0.19	17.0 ± 0.7	250 ± 11	781 ± 37
PTFE/25GF	1291 ± 61	2.46 ± 0.11	3.65 ± 0.17	15.0 ± 0.7	200 ± 8	1291 ± 61
PTFE/25C	952 ± 42	3.22 ± 0.14	5.6 ± 0.25	12.0 ± 0.5	100 ± 4	952 ± 42
PTFE/15G	802 ± 38	3.17 ± 0.13	5.25 ± 0.23	13.0 ± 0.5	200 ± 9	802 ± 38

**Table 3 materials-14-01712-t003:** Stereological analysis results for PTFE and its composites: the number of particles and Influence zones.

Type	The Number of Particles (1/mm^2^)			Influence Zones (µm^2^)		
	Before Test	Static Compression Test	Dynamic Compression Test	Before Test	Static Compression Test	Dynamic Compression Test
PTFE/30B	2167	1993	2192	315	385	289
PTFE/40B	3097	3297	3836	169	227	183
PTFE/60B	4368	5226	5552	102	86	90
PTFE/15GF	184	148	198	3328	4525	3401
PTFE/25GF	240	331	293	2762	1786	1992
PTFE/25C	1672	2064	1800	218	147	189
PTFE/15G	1326	2301	1600	269	125	227

## Data Availability

The data presented in this study are available on request from the corresponding author. The data are not publicly available due to technical limitations. Specific (or example) data may be sent on request by the corresponding author.

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
