# Peer review of "Mechanical Behavior and Morphological Study of Polytetrafluoroethylene (PTFE) Composites under Static and Cyclic Loading Condition"

_materials, 2021, doi:10.3390/ma14071712_

Round 1
Reviewer 1 Report
Interesting results are reported and the manuscript was reviewed for publication in Materials-MDPI Journal. However, to improve the article further, the authors can consider following major points; -
- The abstract is well written. However, what is the target application of the work need to be included in last line of the abstract. Moreover, the abstract lacks the discussion on different filler geometry as claimed in title.
- The idea of using bronze and other mineral filler as reinforcing agent is interesting. However, why authors use term “filler geometry” in the title is confusing? Authors did not discuss this factor (filler geometry) in introduction of the paper? So, please cite more papers from literature which include use of the fillers used in this work and those discuss role of filler geometry on properties of the composites. Few papers citable are [https://doi.org/10.1016/j.sna.2019.111712],
[https://doi.org/10.1002/pc.24692], [https://doi.org/10.1016/j.compscitech.2017.01.023].
- Are these fillers have different geometry? Different shape? Different size or different surface area? If yes, why not to describe this part in experimental section and correlate them with the properties and fatigue tests performed in results and discussions section of the article?
- The table of compound formulation is missing? Moreover, what about outline schematic diagram of the preparation of composites?
- In table-1, it was claimed by the authors that with addition of fillers, the fracture strain in all the filled composites is lower than the unfilled specimen. Please discuss this point in detail. Moreover, while the tensile strength improves only in case bronze, why then authors categories glass fiber and graphite power as reinforcing fillers? This is highly confusing?
- Page 5 #line 211, its “graphite particles” and NOT “grapgite particles”. Please crosscheck the typo.
- In Figure 2, how the dissipation energy is estimated? Why the dissipation losses is negative in case of glass fiber reinforced composites? It must be related to poor dispersion of glass fibers, poor interfacial interaction between filler and matrix and poor stress transfer from matrix to fillers aggregates? Please clarify and discuss on these aspects?
- Page 7 #line 272-273, authors claim “the addition of bronze powder only slightly changed the crystallographic structure of PTFE”, However, the crystallography of the composites are generally studied through XRD. Please clarify?
- Figure 11 must be shifted to Figure 1. Most necessarily, SEM or TEM of the fillers morphology and discussion on their geometry (as mentioned in title of paper) must be discussed before composite discussion section.
- In Figure 12, how authors determine shape coefficients at different conditions?
- Conclusion need to be improved. Authors must discuss how geometry of the filler affects the properties of the composites? How much quantity and quality of the properties improved with adding fillers and so on?
Author Response
We would like to thank the Reviewer for the helpful suggestions and the critical comments, which helped us to improve the message of our manuscript. The answers to the comments are listed below and the respective changes in the revised text are marked red.
- The abstract is well written. However, what is the target application of the work need to be included in last line of the abstract. Moreover, the abstract lacks the discussion on different filler geometry as claimed in title.
Answer: The abstract was rewritten and information about potential applications and discussion of filler geometry has been added. In addition, we modified the title of the manuscript: “Mechanical behavior and morphological study of polytetrafluoroethylene (PTFE) composites under static and cyclic loading condition”.
“The key goal of this study was to characterize polytetrafluoroethylene (PTFE) based composites with the addition of bronze particles and mineral fibers/particles. The addition of individual fillers was as follows: bronze – 30-60 wt.%, glass fibers – 15-25 wt.%, coke flakes – 25 wt.% and graphite particles – 15 wt.%. Both static and dynamic tests were performed and the obtained results were compared with the microscopic structure of the obtained fractures. The research showed that the addition of 60 wt.% bronze and other mineral fillers improved the values obtained in the static compression test to 60% in the case of composites with 25 wt.% glass fibers. Fatigue tests have been performed for the compression-compression load up to 100,000 cycles. All tested composites show a significant increase in the modulus as compared to the values obtained in the static compression test. The highest increase in the modulus in the dynamic test was obtained for composites with 25 wt.% of glass fibers (increase by 85%) and 25 wt.% of coke flakes (increase by 77%), while the lowest result was obtained for the lowest content of bronze particles (decrease by 8%). Dynamic tests have shown that composites with “semi-spherical” particles are characterized by the longest service life and a slower fatigue crack propagation rate than in the case of the long glass fibers. Studies have shown that particles with smaller sizes and more spherical shape have a higher ability to dissipate mechanical energy, which allows their use in friction nodes. On the other hand, composites with glass fiber and graphite particles can be successfully used in applications requiring high stiffness with low amplitude vibrations.”
- The idea of using bronze and other mineral filler as reinforcing agent is interesting. However, why authors use term “filler geometry” in the title is confusing? Authors did not discuss this factor (filler geometry) in introduction of the paper? So, please cite more papers from literature which include use of the fillers used in this work and those discuss role of filler geometry on properties of the composites. Few papers citable are [https://doi.org/10.1016/j.sna.2019.111712], [https://doi.org/10.1002/pc.24692], [https://doi.org/10.1016/j.compscitech.2017.01.023].
Answer: Thank you for recommendations, we added extra paragraph about filler geometry.
“In the case of composites reinforced with particles/fibers, not only their initial mechanical, thermal or physical properties are important, but also their morphology is often important for the initial properties of polymer composites [14]. One of the methods of assessing the influence of particle geometry is the finite element method. This method was used by Qing to investigate the effects of particle size, locations and orientations on mechanical properties of metal-matrix composites [15]. SiC particles of various shapes were used as the filler: circular, octagonal, hexagonal and square. As shown by the tensile failure stresses studies, they are dependent on the structure of the filler, and the best results were obtained in composites with circular particles, and the lowest with square ones. Another work showing the influence of particle geometry on the properties of composites was Lebar et. al [16]. In their work, they investigated the influence of the particle shape on the spall strength of polyurethane composites reinforced with aluminum oxide. As the research showed, not the shape of the particles but the particle/matrix adhesion had a decisive influence on the properties. Kumar et. al investigated the influence of the shape of carbon nanofillers (CB, CNT I GR) on the mechanical and electro-mechanical properties of resin-based composites (RTV silicone) [17]. The particles were characterized by different morphology: CBs had an oval shape and formed agglomerations, CNTs were characterized by an elongated shape with a large specific surface area, while GR was similar to platelets, but agglomerated to a lesser extent than CB. The research showed that the highest properties were characterized by composites reinforced with elongated fillers (CNT) and the increase in Young's modulus was 350%, while the lowest values were obtained for composites with CB (increase by 130%). This was most likely due to particle agglomeration and the inability to transfer the load from the die to the filler.”
- Are these fillers have different geometry? Different shape? Different size or different surface area? If yes, why not to describe this part in experimental section and correlate them with the properties and fatigue tests performed in results and discussions section of the article?
Answer: We added extra section: 3.1. Characteristics of fillers
“Figure 2 presents images showing the morphology of particles used as fillers in the produced composites. As can be seen, the particles have different structures. Bronze particles have a globular structure of 20-30 µm, and their surface is not as developed as in the case of coke and graphite particles. The coke and graphite particles have a developed lamellar structure of quite irregular shape. The graphite particles are much larger and much more elongated than the coke particles. In contrast, glass fibers have the typical elongated structure characteristic of fibers. The diameter of the glass fibers is 11 µm and the length is max. 200 µm.”
- The table of compound formulation is missing? Moreover, what about outline schematic diagram of the preparation of composites?
Answer: Thank you for this idea. We added table with compound formulations and schematic diagram of the preparation of composites
Table 1. Abbreviations and compositions of produced composites
Abbreviation |
Composition |
PTFE |
100 wt.% polytetrafluoroethylene |
PTFE/30B |
70 wt.% polytetrafluoroethylene+ 30 wt.% (11 vol.%) bronze particles |
PTFE/40B |
60 wt.% polytetrafluoroethylene + 40 wt.% (16 vol.%) bronze particles |
PTFE/60B |
40 wt.% polytetrafluoroethylene + 60 wt.% (29 vol.%) bronze particles |
PTFE/15GF |
85 wt.% polytetrafluoroethylene + 15 wt.% (14 vol.%) glass fibers |
PTFE/25GF |
75 wt.% polytetrafluoroethylene + 25 wt.% (23 vol.%) glass fibers |
PTFE/25C |
75 wt.% polytetrafluoroethylene + 25 wt.% (29 vol.%) coke flakes |
PTFE/15G |
85 wt.% polytetrafluoroethylene + 15 wt.% (15 vol.%) graphite particles |
Figure 1. A schematic diagram of the manufacturing
- In table-1, it was claimed by the authors that with addition of fillers, the fracture strain in all the filled composites is lower than the unfilled specimen. Please discuss this point in detail. Moreover, while the tensile strength improves only in case bronze, why then authors categories glass fiber and graphite power as reinforcing fillers? This is highly confusing?
Answer: We apologize for this mistake, of course it was an increase in stiffness for all composites. In addition, more details were added about fracture strain.
“The higher the filler content, the lower the deformability. This relationship indicates the formation of discontinuities in the matrix, which reduce the plastic deformation capacity of the tested composites. The addition of particles resulted in the creation of more places where breaking stress could occur. The higher the volume of a given filler, the higher the number of particles and thus the more destructive places. These studies also confirm this relationship: PTFE/25C (29 vol.% – 100%)< PTFE/60B (29 vol.% – 181%)< PTFE/25GF (23 vol.% – 200%)<PTFE/25C (15 vol.% – 200%)< PTFE/15GF (14 vol.% – 250%)< PTFE/40B (16 vol.% – 367%)< PTFE/40B (11 vol.% – 383%).”
- Page 5 #line 211, its “graphite particles” and NOT “grapgite particles”. Please crosscheck the typo.
Answer: Thank you for this notice, we corrected this mistake. We checked the manuscript for language and corrected them accordingly.
- In Figure 2, how the dissipation energy is estimated? Why the dissipation losses is negative in case of glass fiber reinforced composites? It must be related to poor dispersion of glass fibers, poor interfacial interaction between filler and matrix and poor stress transfer from matrix to fillers aggregates? Please clarify and discuss on these aspects?
Answer: The dissipation energy was calculated on the basis of the surface area of the hysteresis loops obtained in one complete strain cycle. The Mathcad Prime 6.0.0.0 program was used for the calculation.
Due to the high stiffness of the glass fiber, it is difficult to expect a significant improvement in the ability to dissipate energy by PTFE composites with glass fiber (lower viscoelastic deformation ability). In addition, the adhesion between the fiber and the matrix was not sufficient and the fibers became drawn under cyclic loads.
- Page 7 #line 272-273, authors claim “the addition of bronze powder only slightly changed the crystallographic structure of PTFE”, However, the crystallography of the composites are generally studied through XRD. Please clarify?
Answer: Thank you for this comment and we agree with the Reviewer, however, in our work we only described the change in the crystalline structure in terms of quantity and not qualitative. Due to the lack of availability of XRD measurements, we have described the SEM photos, which show the fragmentation of the crystallographic structure (smaller spheresolites). Of course, in the future, as soon as the research is available, the crystallographic structure will be assessed by means of XRD.
- Figure 11 must be shifted to Figure 1. Most necessarily, SEM or TEM of the fillers morphology and discussion on their geometry (as mentioned in title of paper) must be discussed before composite discussion section.
Answer: We moved Figure 11 as Figure 1 and we added discussion about fillers.
- In Figure 12, how authors determine shape coefficients at different conditions?
Answer: The shape coefficient (aspect ratio) is defined by the ratio of the F(90) to the F(0). Due to the lack of differences in the method of measurement, the shape coefficient calculated on the basis of transverse measurements was provided for the analysis.
- Conclusion need to be improved. Authors must discuss how geometry of the filler affects the properties of the composites? How much quantity and quality of the properties improved with adding fillers and so on
Answer: The conclusion has been rewritten.

Reviewer 2 Report
The authors of this work, “Effect of fillers geometry on static and dynamic mechanical properties of composites based on polytetrafluoroethylene (PTFE)”, have presented a detailed experimental results analysis for the modification of PTTE with different materials. it is an excellent manuscript.
- The manuscript presents a detailed technique used to produce the composite materials and the detailed analysis for characterizing them.
- The strength of the composite material is classified using the general mechanical properties.
- The effect of the shape and size of the particles were also used to support their argument.
- The paper is well organized and structured.
- The manuscript shows that the application of PTTE can be extended to friction parts if the strength is modified with some materials.
- The evidence and argument support the conclusion presented.
Few comments to address:
- The authors should read the manuscript again to correct some statements that are not clear and some grammatical errors such “we can more and more often find”, “in world literature.”
- Please rephrase this statement “Towards to exclude errors in measurements and to prepare static data, at least three samples were tested repeatedly.”
- The author should describe how the “shape coefficient” for different materials is measured.
- The x-axis legend is missing in Fig 13.
- The author explains in the introduction that “An analysis of the fillers was performed, which determined the following parameters: volume fraction of particles in the matrix, particle surface area and Feret diameters F (0) and F (90) based on which the shape coefficient was calculated”. However, the author does not consider the volume fraction of the particles in the discussion. Please give a brief description of the effect of filler volume fraction on the strength of the material.
I recommend the publication of this article after addressing the comments.
- The authors should read the manuscript again to correct some statements are not clear to understand and some grammatical errors such “we can more and more often find”, “in world literature.”
- Please rephrase this statement “Towards to exclude errors in measurements and to prepare static data, at least three samples were tested repeatedly.”
Author Response
We would like to thank the Reviewer for overall positive evaluation of our manuscript and for helpful suggestions. The answers to the comments are listed below and the respective changes in the revised text are marked red.
The authors should read the manuscript again to correct some statements that are not clear and some grammatical errors such “we can more and more often find”, “in world literature.”
Answer: We checked the manuscript for language and corrected them accordingly.
- Please rephrase this statement “Towards to exclude errors in measurements and to prepare static data, at least three samples were tested repeatedly.”
Answer: This phrase has been modified:
“To exclude errors and create statistical analysis at least three samples were tested for each test and for each produced material.”
The author should describe how the “shape coefficient” for different materials is measured.
Answer: The shape coefficient (aspect ratio) is defined by the ratio of the F(90) to the F(0). Due to the lack of differences in the method of measurement, the shape coefficient calculated on the basis of transverse measurements was provided for the analysis.
- The x-axis legend is missing in Fig 13.
Answer: Thank you for this notice, the x-line has been added to the chart
- The author explains in the introduction that “An analysis of the fillers was performed, which determined the following parameters: volume fraction of particles in the matrix, particle surface area and Feret diameters F (0) and F (90) based on which the shape coefficient was calculated”. However, the author does not consider the volume fraction of the particles in the discussion. Please give a brief description of the effect of filler volume fraction on the strength of the material.
Answer: We added information about effect of filler volume fraction on the strength of the material.
“With the increase in the volume of the filler, the Young's modulus under compression increases, the highest increase was recorded for PTFE/25GF (23 vol.%) – 69%, and the lowest for PTFE/30B (11 vol.%) – a decrease by 30%. Due to the high stiffness and larger dimensions of glass fiber to bronze and coke particles, despite the higher content (29 vol.%), glass fiber composites have a higher Young's modulus due to the above-mentioned aspects.”
“The higher the volume of a given filler, the higher the number of particles and thus the more destructive places. These studies also confirm this relationship: PTFE/25C (29 vol.% – 100%) < PTFE/60B (29 vol.% – 181%)< PTFE/25GF (23 vol.% – 200%)<PTFE/25C (15 vol.% – 200%)< PTFE/15GF (14 vol.% – 250%)< PTFE/40B (16 vol.% – 367%)< PTFE/40B (11 vol.% – 383%).”

Reviewer 3 Report
The main aim of this publication was to characterize polytetrafluoroethylene (PTFE) based composites with the addition of bronze particles (30-60 wt.%) glass fibers (15-25 wt.%), coke flakes (25 wt.%) and graphite particles ( 15 wt.%). For characterization of the prepared composites static and dynamic mechanical tests were performed and SEM measurements also were done. The main goal of the research was to open a wider field of application of the PTFE composites. Coal and bronze fillers have been shown to contribute to energy dissipation. Research, of course, has a scientific antecedents, but in such a generalization we can see a relatively new result. The conclusions are in line with the results. I do not see wishfull thinking in the results.
The publication is well written, easy to understand, and does not contain complex linguistic structures. There are basically no typographical errors in the publication, but the notations a, b, c, and d in Figure 1 are missing.
The characterization of the fillers is very incomplete. The average particle size, size distribution, etc. are not given.
PTFE is a very stable compound, it hardly reacts with any compound, however, its surface is also very non-polar, so the compatibilizer would have been useful for polar fillers.
Author Response
We would like to thank the Reviewer for the helpful suggestions and the critical comments, which helped us to improve the message of our manuscript. The answers to the comments are listed below and the respective changes in the revised text are marked red.
- The publication has basically no typing errors, but in Figure 1, the notations a, b, c and d are missing.
Answer: Thank you for this notice, the missing letters have been added to the chart.
- The characterization of the fillers is very incomplete. The average particle size, size distribution, etc. are not given.
Answer: We added extra section: 3.1. Characteristics of fillers
“Figure 2 presents images showing the morphology of particles used as fillers in the produced composites. As can be seen, the particles have different structures. Bronze particles have a globular structure of 20-30 µm, and their surface is not as developed as in the case of coke and graphite particles. The coke and graphite particles have a developed lamellar structure of quite irregular shape. The graphite particles are much larger and much more elongated than the coke particles. In contrast, glass fibers have the typical elongated structure characteristic of fibers. The diameter of the glass fibers is 11 µm and the length is max. 200 µm.”
- PTFE is a very stable compound, it hardly reacts with any compound and in the same time the surface of it is very apolar, so a compatibilizer would have been useful for polar fillers.
Answer: This is a very valid point. As the authors themselves mention that the modification of particles could contribute to an increase in the adhesion of the filler to the matrix:
“SEM microscopic images showed that insufficient fiber/matrix adhesion was observed, which contributed to a reduction in the properties obtained during the tensile test. In order to improve the results obtained in this work in the future, the surfaces of the particles should be modified or their fragmentation increased. Additionally, a compatibilizer may be considered to increase the miscibility of the apolar PTFE with the fillers.”
Future work will focus on increasing the particle compatibility of PTFE

Round 2
Reviewer 1 Report
I have few comments before acceptance.
Comments -
- In introduction, #line 118, there is a typo, its (s (CB, CNT, GR) and NOT (CB, CNT I GR).
- In table#2, table content is missing
Author Response
Dear Reviewer,
Thank you for your comments, we improved all of mentioned errors